# Effects of Law Enforcement Load Carriage Systems on Muscle Activity and Coordination during Walking: An Exploratory Study

**DOI:** 10.3390/s23084052

**Published:** 2023-04-17

**Authors:** Joel Martin, Megan Sax van der Weyden, Marcie Fyock-Martin

**Affiliations:** Sports Medicine Assessment Research & Testing (SMART) Laboratory, George Mason University, Manassas, VA 20110, USA; msaxvand@gmu.edu (M.S.v.d.W.); mfyock@gmu.edu (M.F.-M.)

**Keywords:** lower back, injury, police, military, firefighter

## Abstract

Law enforcement officers (LEOs) commonly wear a duty belt (DB) or tactical vest (TV) and from prior findings, these forms of load carriage (LC) likely alter muscular activity. However, studies on the effects of LEO LC on muscular activity and coordination are limited in the current literature. The present study examined the effects of LEO load carriage on muscular activity and coordination. Twenty-four volunteers participated in the study (male = 13, age = 24.5 ± 6.0 years). Surface electromyography (sEMG) sensors were placed on the vastus lateralis, biceps femoris, multifidus, and lower rectus abdominus. Participants completed treadmill walking for two load carriage conditions (duty belt and tactical vest) and a control condition. Mean activity, sample entropy and Pearson correlation coefficients were computed for each muscle pair during the trials. The duty belt and tactical vest resulted in an increase in muscle activity in several muscles; however, no differences between the duty belt and tactical vest were found. Consistently across the conditions, the largest correlations were observed between the left and right multifidus (r = 0.33–0.68) and rectus abdominus muscles (0.34–0.55). There were statistically small effects (*p* < 0.05, η^2^ = 0.031 to 0.076) of the LC on intermuscular coordination. No effect (*p* > 0.05) of the LC on sample entropy was found for any muscle. The findings indicate that LEO LC causes small differences in muscular activity and coordination during walking. Future research should incorporate heavier loads and longer durations.

## 1. Introduction

Law enforcement is a physically demanding occupation [1] and often results in high rates of musculoskeletal injury due to the use of protective gear, physical confrontations with suspects or long periods remaining in stationary positions [2,3]. Specifically, low back pain (LBP) is highly prevalent in law enforcement officers (LEOs) and has been associated with both physical and emotional consequences. [4,5,6,7]. These include reduced overall wellbeing [8], job time loss [8,9], forced separation from the department for medical reasons [8,9,10] and the potential for unsuccessful recovery. The financial costs for taxpayers when a law enforcement officer is on leave from their job due to injury (i.e., LBP) is estimated to be in the billions annually [11]. Given the negative consequences of LEO LBP, understanding the causes of LBP amongst LEOs is vital to address the issue. 

Generally, musculoskeletal pain is known to alter common movement patterns, such as gait, which is readily observable [12]. However, numerous studies have reported subsequent alterations to the motor control of muscles in the presence of chronic musculoskeletal pain [13]. It has been reported in individuals with LBP that there are alterations to muscle activity patterns during both standing [14] and walking [15]. Silfies et al. used surface electromyography (sEMG) to compare activation levels of five trunk muscles in 20 participants with chronic LBP to 20 asymptomatic controls [14]. Higher levels of abdominal muscle activation with lower ratios of activation between synergistic abdominal muscles were found in the chronic LBP group. Interestingly, no differences in muscle activity of the erector spinae were found between the LBP and asymptomatic groups. Similarly, Lamoth and colleagues examined the muscle activity patterns of 19 participants with chronic LBP and 14 asymptomatic controls during treadmill walking [15]. Higher levels of muscle activity in the erector spinae for the LBP group compared to the control group during walking were reported. Additional analyses found altered muscle coordination patterns between the LBP and control group, indicating reduced global intermuscular coordination. The aforementioned studies [14,15] provide support for the observation that alterations in muscle activity levels can be indicative of LBP. 

Law enforcement officers utilize load carriage (LC) systems (i.e., duty belt, tactical vest), which they are typically required to wear while on duty [10,16]. LC is well documented to alter physiological, kinematic, and kinetic measures during walking [17,18]; however, few studies have examined the effects LC has on muscle activity during walking tasks [19,20,21]. Several studies reported the effects of LEO LC on muscle activity during quiet standing [22,23]. In a sample of college students, Motmans et al. found that backpacks loaded with 15% body mass reduced erector spinae but increased rectus abdominus activity [22]. Interestingly, the LC also increased rectus abdominus activation asymmetry [22]. Similarly, Park and colleagues reported that various configurations of LC increased muscle activity with a greater load and that an uneven weight distribution of load had a greater effect on muscle activity [23]. 

Surprisingly, only a few studies have reported on the effects of LC during walking; however, a consistent finding has been an increase in muscle activity due to LC [19,20,21]. These studies have included samples of female hikers [20], marine recruits [19], and healthy middle-age adults [21], which could be comparable to LEO populations. However, none of the studies reporting on the effects of LC on muscle activity during walking have recorded the effects on abdominal or low back musculature [19,20,21]. Rather, all the studies considered the effects on lower extremity muscles [19,20,21], which could be expected to increase activities due to additional force production requirements as a byproduct of LC mass.

Several limitations in the current literature regarding the effects of LEO LC systems on muscle activity presently exist. To our knowledge, a direct comparison of a duty belt to a tactical vest on muscle activity during walking has not been reported. Secondly, the current literature has most commonly reported that muscle activation levels are the primary outcome with inconsistent findings [22,23]. Non-linear analyses, such as sample entropy, have been shown to be more sensitive to subtle changes in biomechanical outcomes [24] and have shown promise for the analysis of sEMG data [25,26]. Additionally, given that LBP is attributed to lumbar spine instability [27], examining alterations in intermuscular coordination due to acute LC exposure would add to the existing literature [28]. 

The purpose of the present study was to examine the effects of LEO LC systems on core muscle activity patterns during gait. Specifically, we aimed to compare muscle activity levels and intermuscular coordination patterns between (1) a control condition (no load) and (2) different types of LC systems (duty belt vs. tactical vest). Alterations in muscle activity patterns would provide further insight into the contributions of LC to LBP and could potentially provide support for one type of LC carriage system over another. We hypothesized that (1) both types of LC system would lead to significant changes and (2) there would be no differences between equal mass LC systems in muscle activation levels and intermuscular coordination during gait.

## 2. Materials and Methods

### 2.1. Study Design and Participants

A repeated measures randomized crossover design was used that required participants to attend a single data collection session. Pre-testing instructions were to avoid vigorous exercise prior to the testing session. The eligibility criteria were: 18–45 years of age, a body mass index (BMI) less than 30 m/kg^2^, to be physically active, with no prior history of lower extremity injury or LBP in the last 6 months or the incapacity to perform a barbell deadlift equivalent to their body mass. The study was approved by the George Mason University Institutional Review Board (IRB approval #: 1455213-1). Prior to participation, an informed consent form was signed, and participants were screened for eligibility. Data were collected from 24 non-LEO participants (13 male, age: 24.5 ± 6.0 years, height: 169.3 ± 9.8 cm, mass: 73.0 ± 11.1 kg, BMI: 25.4 ± 2.4 kg/m^2^). 

### 2.2. Procedures

The present study analyzed a data set as part of a larger project examining the effects of LEO LC on human performance. After the screening was completed and the informed consent was obtained, participants answered a battery of questionnaires regarding physical activity, sleep, feelings of energy and fatigue, and moods (Qualtrics, XM, Provo, UT, USA). The questionnaires took approximately 15 to 20 min to complete. Participants were instrumented with surface electromyography (sEMG) sensors (Trigno, Delsys, Natick, MA, USA) and then completed a battery of cognitive, static, and dynamic movement tasks under 3 conditions: (1) no load, (2) leather LEO belt with a total mass of 7.2 kg and (3) a 7.2 kg LEO tactical style weight vest (Figure 1). The conditions were performed in a random order with the battery of cognitive, static and dynamic tasks always performed in a fixed order. All participants were given 5 min of rest between conditions. The time taken to complete the study was approximately 120 min, with the initial set-up taking 30 min and each experimental condition typically lasting for 20–25 min. For the aim of the present study, the gait task will be the focus of the subsequent sections. 

A stadiometer (Detecto, Webb City, MO, USA) and a digital scale (EatSmart, Tokyo, Japan) were used to measure each participant’s height and mass, respectively. The sEMG sensors were placed bilaterally on the muscles of the trunk and upper thigh segment (Figure 2): multifidus (MF), rectus femoris (RF), biceps femoris (BF), and rectus abdominus (RA). Sensor placement was based on standard guidelines [29]. The sEMG sensors were secured with elastic tape to prevent movement of the sensors during the data collection. All electronic devices (i.e., cell phones, smart watches, etc.) were turned off and/or removed from the testing area. Following the sensor placement, all participants performed a standing warm-up of 10 bird-dogs, 5 inchworms, 12 bodyweight squats, and 12 bodyweight Romanian deadlifts (RDL) for 2 rounds. Immediately after the warm-up, participants performed a series of tasks for each of three conditions in the predetermined randomized condition order. The gait task was performed on a treadmill (Star Trac, Core Health & Fitness, Vancouver, WA, USA) where participants walked at 1.34 m/s and at a 1.5% incline for a 60 s period. EMG activity data were collected for 30 s once the treadmill reached the required velocity and incline.

The sEMG signals from each muscle were recorded at a sampling rate of 2000 Hz. Following data collection, sEMG data were full-wave rectified and bandpass filtered (20 to 490 Hz) with a 4th order Butterworth filter. The rectified sEMG data were then smoothed using a 5 Hz low pass 4th order Butterworth filter. All sEMG filtering and pre-processing was performed in MATLAB (MATLAB 2020a, MathWorks Inc., Natick, MA, USA). Representative processed sEMG data from a single participant are shown in Figure 3.

### 2.3. Statistical Analysis

Outcome measures were initially assessed for normality and extreme values. This was done by condition and not across conditions. Extreme values were defined as a data point that fell more than 3 standard deviations outside the mean [30]. To remove these extreme values and minimize data loss, each variable was independently winsorized to the 1st or 99th percentiles [31]. The level of sEMG activity for the LEO_DB_ and vest conditions was then normalized to the control condition. Lindner and colleagues previously used a similar normalization procedure [32]. Pearson correlation coefficients were computed from the processed sEMG data for each pair of muscles from every trial collected. The sample entropy of each muscle sEMG signal was computed for all trials. For the sample entropy, parameters for m and tolerance r were 2 and 0.2, respectively. These parameter choices have been commonly used for sEMG [25,26] and other biomechanical measures previously [24]. 

Descriptive statistics including the median and interquartile range were computed for sEMG, and the mean and standard deviation were computed for the correlations between muscles based on normality testing. The Pearson correlations were first z-transformed, the descriptive statistics were computed, and then the inverse z-transform was applied to produce the values reported in the results. Based on the normality results, appropriate repeated measures inferential testing was performed to assess the main effect of each condition (3 levels: control, leather LEO_DB_, and vest) [33]. For normally distributed data, the main effects were assessed using a repeated measures analysis of variance (ANOVA) and post hoc comparisons were conducted via pairwise t-tests with a Bonferroni correction. For non-normally distributed data, the main effects were assessed using Friedman’s tests and post hoc comparisons via the Wilcoxon signed ranks test with a Bonferroni correction [34]. Given the sample size of 24, both the Wilcoxon test statistic (W) and the z-statistic (z) are reported. Effect sizes for the repeated measure ANOVAs and Friedman’s tests are reported as partial eta squared (small effect: η^2^ = 0.01–0.06, medium effect: η^2^ => 0.06–0.14, large effect: η^2^ => 0.14) and Kendall’s W (W_k_) (small effect: W = 0.1–0.3, medium effect: W => 0.3–0.5, large effect: W => 0.5) [35]. The Glass rank-biserial correlation coefficient effect sizes were computed. The strength of association for the correlations were assessed as follows: trivial, r = 0 < 0.10; weak, r = 0.10–0.40; moderate, r = 0.41–0.70; strong, r > 0.71 [35]. The statistical analyses were conducted in R (R-Studios, version 2022.2.0, Vienna, Austria). Statistical significance was set to α ≤ 0.05.

## 3. Results

### 3.1. sEMG Amplitude and Sample Entropy

There was a significant main effect of the condition in the left RF (χ^2^(2) = 16.3, *p* = 0.001, W_k_ = 0.34), right RF (χ^2^(2) = 9.0, *p* = 0.011, W_k_ = 0.19), left MF (χ^2^(2) = 8.58, *p* = 0.014, W_k_ = 0.18), and right MF (χ^2^(2) = 13.1, *p* = 0.001, W_k_ = 0.27). Post hoc analysis revealed that the belt resulted in significantly more muscular activity than the control condition in the left RF (W = 38, z = −3.2, *p* = 0.004, rg = 0.653), right RF (W = 49, z = −2.88, *p* = 0.012, rg = 0.589), and left MF (W = 44, z = −3.03, *p* = 0.007, rg = 0.618; Figure 4). The vest resulted in significantly more muscular activity than the control condition in the left RF (W = 25, z = −3.57, *p* = 0.001, rg = 0.729), right RF (W = 66, z = −2.4, *p* = 0.049, rg = 0.490), left MF (W = 64, z = −2.46, *p* = 0.042, rg = 0.502), and right MF (W = 23, z = −3.63, *p* < 0.001, rg = 0.741; Figure 1). The belt and vest conditions were not significantly different from one another. Regarding sample entropy, no main effects of the conditions were found (Table 1).

### 3.2. Intermuscular Coordination

Consistently across conditions for walking, the largest correlations were observed between the left and right MF and RA (Table 2). There was a main effect of condition on the correlation between left RF with left AB (F(2,46) = 5.487, *p* = 0.007, η^2^ = 0.076; post hoc: BELT < CONTROL, *p* = 0.021), left BF with right RA (χ^2^(2) = 6.083, *p* = 0.048, W_k_ = 0.13; post hoc: CONTROL > VEST, *p* = 0.009), right BF with left RA (F(2,46) = 3.582, *p* = 0.036, η^2^ = 0.038; no significant post hoc comparisons), right BF with right RA (χ^2^(2) = 8.333, *p* = 0.016, W_k_ = 0.17; post hoc: VEST > CONTROL, *p* = 0.002), and right BF with left MF (F(2,46) = 3.226, *p* = 0.049, η^2^ = 0.031; no significant post hoc comparisons). 

## 4. Discussion

The aim of the study was to investigate the effects of LEO-style LC on core muscle activity and coordination during walking, which is a common activity performed by LEOs. Our first hypothesis, that both types of load carriage would result in changes in these measures as compared to unloaded walking, was supported by the findings. The second hypothesis, that there would be no differences between types of LC, was supported as well. The finding that a short exposure to relatively light LEO-style LC alters the neuromuscular activity of lower extremity and core muscles has several important practical implications. Additionally, several avenues of future research are advised based on the results of this exploratory study and these are discussed below. 

As mentioned previously, increased muscle activity in lower extremity musculature during walking with LC has been previously studied [19,20,21]; however, the effects of LEO LC on core musculature during walking are novel findings. The increased levels of muscle activity in the multifidus are notable considering that higher levels of core muscle activity in patients with LBP, as compared to healthy controls, have been reported previously [14,15]. A common consequence of LC is that individuals increase their forward lean [36,37] and anteriorly tilt the pelvis [38,39]. These changes in posture would subsequently increase the length of low back musculature and necessitate greater force production to maintain posture as the moment about the low back is increased [17]. Although we did not record trunk lean, the increase in the MF muscle activity in our study is likely related to increased demands on low back musculature to maintain an upright trunk posture. It should be noted that the magnitudes of loads and duration of activity in our study were less than those used in many prior studies [36,37,40]; yet, increased low back muscular activity was observed. 

A strength of the current study is the investigation into the effects of LC on intermuscular coordination. In order to perform dynamic movements (i.e., walking), muscles of the body must interact to produce coordinated movements [41,42]. Specifically, muscles must interact to produce net joint movement as well as stabilization [42]. Core stability, in particular, requires coordination of the muscles on the anterior-posterior and lateral regions of the core region of the body to stabilize during flexion/extension, lateral bending and rotation movements [43]. To our knowledge, this is one of the first studies to report on the effects of LC on muscular coordination via sEMG sensors, whereas much of the prior literature has focused on changes in single muscle amplitude due to various types of load while walking [19,20,21]. We found the strongest associations to be between abdominals and lower back musculature bilaterally, which LC did not affect (Table 2). There were effects of LC on the coordination of several muscle pairs due to load. Interestingly, all these instances were between a muscle of the thigh segment and the core. The importance of core stability for mitigating the negative effects of non-specific LBP is the focus of a systematic review by Frizziero and colleagues [44]. 

An important issue to consider is that the core musculature does not act only to stabilize the body in flexion and extension but also rotationally. Moreover, trunk rotation is a known risk factor for LBP [45]. During gait that features alternating rhythmic motion of the lower body segments, rotational momentum is generated about the lumber spine [42]. In order to stabilize rotationally, the contralateral core muscles should contract to the moving limb. For example, during walking, when the left thigh segment moves forward, core musculature on the right side of the body would need to contract in order to stabilize the lumbar spine. Our muscular coordination results indicate moderate associations between the RF and contralateral MF muscles, highlighting the particular importance of these muscle pairs. Considering the previously discussed findings related to changes in MF activity, our findings warrant further investigation into the effects of longer durations, heavier loads, and faster gait speeds on muscular coordination under load. Subsequently, the mechanisms of load carriage on LBP could be better understood.

There were several limitations of the study. One primary limitation was that our sample subjects were not LEOs, and thus the generalizability of the results to LEO populations could be limited. However, our sample would be comparable to recruits entering law enforcement training academies. Law enforcement recruits are often selected from the general population who would have not necessarily been exposed to LC previously. Several studies have well documented that the time spent at LEO academies are a period in the career of a LEO with the highest injury rates [3,46]. Additionally, physical fitness (i.e., body composition, muscular strength, muscular and aerobic endurance) was not controlled for. It has been shown that physical fitness characteristics influence how individuals respond under LC conditions [47,48,49]. Thus, future studies would be advised to include fitness assessments to control for the confounding factor of fitness. A third limitation was the duration of the study and time spent walking. Each walking trial lasted for only 60 s and participants were under load for less than an hour in total (~20 min per condition) and walked for a brief amount of time. Thus, the levels of fatigue experienced were likely low as compared to wearing the same equipment for an 8-hour or more shift as muscular coordination could change over time. Of the studies reporting on the effects of lower extremity muscle activity during prolonged walking with load, the findings are inconsistent. For example, Simpson et al. found that increased walking distance decreased the amplitude of several muscles in the lower extremity [20]. Whereas, Rice and colleagues found no difference in the amplitude of muscle activity [19]. As a note, researchers building on the current findings should assess and quantify muscular fatigue in future studies rather than assuming that because the duration is longer, participants are fatigued. The median frequency of sEMG signals is sensitive to muscle fatigue and would provide insights into the fatigue states of muscles [50]. Examining the effect of LC on muscle activity and coordination during walking over time periods reflective of actual LEO shifts would add to the existing literature. 

## 5. Conclusions

In conclusion, the increased muscle activity due to LEO LC in some, but not all muscles, indicates altered recruitment patterns and intermuscular coordination compared to unloaded walking over short duration time periods. However, there is no clear support for the use of one type of LC over the other. The present findings are interesting considering injury rates [3], specifically those related to the low back in LEO [9,10], combined with the relatively low dose of load carriage exposure in terms of mass and duration used in the present study. Practitioners working with LEOs should be aware that altered muscular activity and coordination between core muscles and primary movers in the lower extremity of the body occurs due to LC, and individuals may benefit from an acclimatization period to avoid injury caused by altered muscular activation patterns. Wearable sEMG sensors provide a viable technique to study the neuromuscular effects of load carriage in laboratory and field settings and could be valuable to screen individuals at higher risk of incurring injury due to altered muscular activity resulting from LC. 

## Figures and Tables

**Figure 1 sensors-23-04052-f001:**
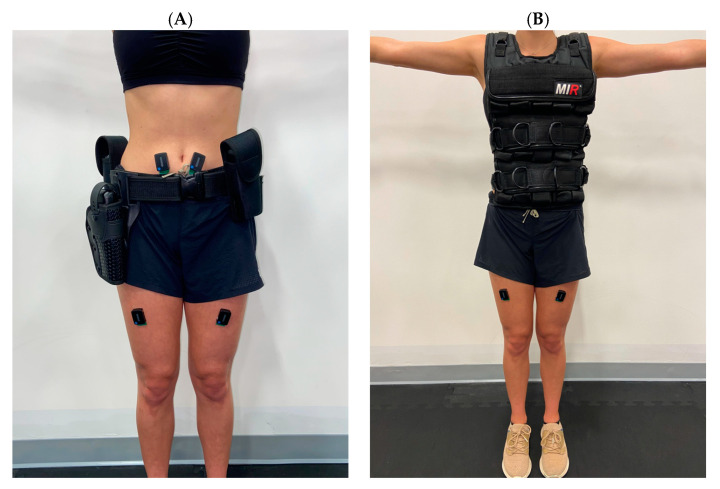
Representative image of participant wearing law enforcement (**A**) duty belt and (**B**) tactical vest.

**Figure 2 sensors-23-04052-f002:**
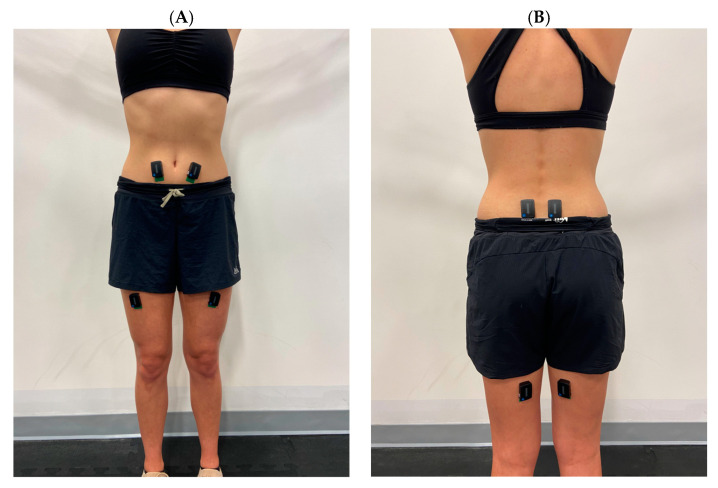
Locations of wearable surface electromyography sensors on the body. Sensors were placed bilaterally on the multifidus, rectus femoris, biceps femoris, and rectus abdominus. Panel (**A**) illustrates sensors placed on muscles of the anterior side of the body. Panel (**B**) illustrates sensors placed on muscles of the posterior side of the body.

**Figure 3 sensors-23-04052-f003:**
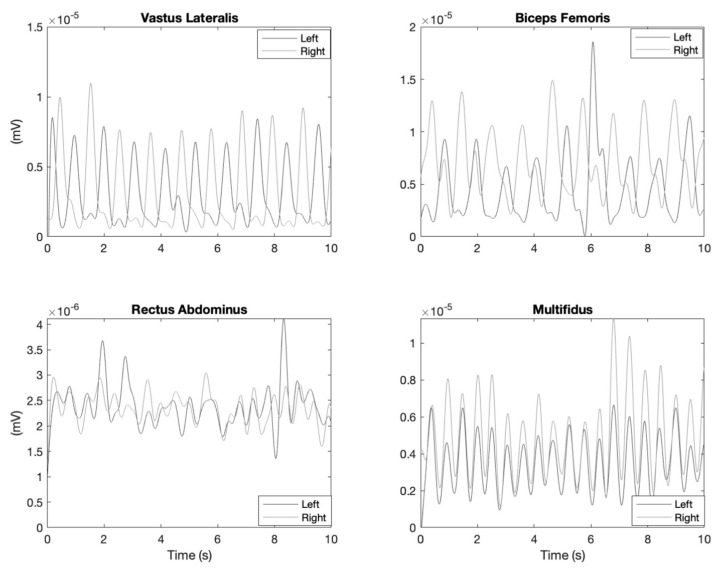
Representative processed surface electromyography from a single subject for a law enforcement belt walking trial. Only 10 of 30 s of collected data are shown to enhance the clarity. During processing, surface electromyography data were full-wave rectified and bandpass filtered (20 to 490 Hz) with a 4th order Butterworth filter. The rectified sEMG data were then smoothed using a 5 Hz low pass 4th order Butterworth filter.

**Figure 4 sensors-23-04052-f004:**
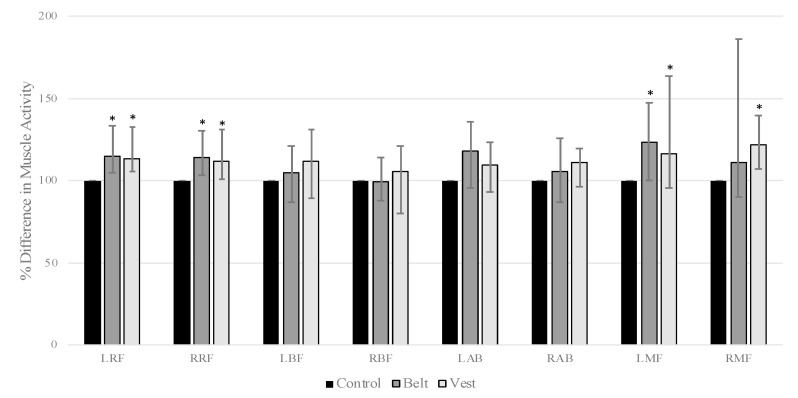
Comparison of muscle activity between conditions while walking. Values are medians and interquartile range in the bracket computed on data from all subjects. * denotes statistical difference compared to control.

**Table 1 sensors-23-04052-t001:** Sample entropy of muscles between conditions while walking.

Muscle	Control	Belt	Vest	*p*-Value
Left Rectus Femoris	0.011 (0.008, 0.013)	0.011 (0.006, 0.013)	0.011 (0.008, 0.13)	0.061
Right Rectus Femoris	0.010 (0.006, 0.013)	0.009 (0.006, 0.013)	0.008 (0.007, 0.012)	0.275
Left Biceps Femoris	0.007 (0.006, 0.008)	0.007 (0.006, 0.009)	0.006 (0.005, 0.008)	0.539
Right Biceps Femoris	0.006 (0.006, 0.008)	0.006 (0.005, 0.007)	0.006 (0.005, 0.007)	0.550
Left Rectus Abdominus	0.012 (0.009, 0.015)	0.013 (0.010, 0.014)	0.012 (0.011, 0.014)	0.959
Right Rectus Abdominus	0.012 (0.010, 0.014)	0.011 (0.010, 0.014)	0.012 (0.010, 0.014)	0.687
Left Multifidus	0.014 (0.013, 0.015)	0.014 (0.014, 0.015)	0.015 (0.012, 0.015)	0.957
Right Multifidus	0.015 (0.014, 0.016)	0.015 (0.014, 0.015)	0.015 (0.013, 0.016)	0.738

Note: None of the muscles were found to have normally distributed sample entropy. Values are the median and interquartile range with *p*-values from Friedman’s test to compare conditions.

**Table 2 sensors-23-04052-t002:** Correlation coefficients of muscles between conditions during walking.

	*Control*		*Belt*		*Vest*			
Muscle Pair	M	SD	M	SD	M	SD	*p*-Value	Post-Hoc
LRF-RRF	0.16	0.43	0.11	0.41	0.14	0.44	0.115	
LRF-LBF	0.50	0.52	0.51	0.55	0.47	0.52	0.755	
LRF-RBF	−0.31	0.29	−0.28	0.25	−0.29	0.32	0.214	
LRF-LAB	0.09	0.19	0.18	0.21	0.23	0.20	0.007	Belt < Control
LRF-RAB	0.03	0.22	0.10	0.23	0.05	0.23	0.847	
LRF-LMF	0.33	0.27	0.32	0.28	0.30	0.31	0.759	
LRF-RMF	0.46	0.30	0.46	0.30	0.46	0.34	0.970	
RRF-LBF	−0.21	0.31	−0.18	0.32	−0.22	0.31	0.883	
RRF-RBF	0.46	0.52	0.42	0.50	0.47	0.56	0.214	
RRF-LAB	0.05	0.19	0.14	0.18	0.05	0.22	0.453	
RRF-RAB	0.08	0.17	0.15	0.23	0.17	0.26	0.175	
RRF-LMF	0.54	0.30	0.50	0.35	0.52	0.35	0.610	
RRF-RMF	0.38	0.22	0.39	0.23	0.39	0.24	0.994	
LBF-RBF	−0.39	0.27	−0.32	0.21	−0.38	0.25	0.185	
LBF-LAB	0.14	0.19	0.18	0.26	0.18	0.18	0.697	
LBF-RAB	0.04	0.14	−0.03	0.15	−0.07	0.19	0.048	Vest < Control
LBF-LMF	0.27	0.26	0.26	0.37	0.20	0.29	0.425	
LBF-RMF	0.29	0.25	0.26	0.27	0.27	0.28	0.697	
RBF-LAB	0.02	0.21	0.01	0.16	−0.06	0.18	0.036	None
RBF-RAB	0.06	0.19	0.11	0.28	0.22	0.28	0.016	Vest > Control
RBF-LMF	0.27	0.30	0.15	0.31	0.27	0.32	0.049	None
RBF-RMF	0.28	0.30	0.20	0.24	0.26	0.30	0.263	
LAB-RAB	0.55	0.34	0.55	0.39	0.50	0.41	0.686	
LAB-LMF	0.17	0.21	0.24	0.20	0.19	0.16	0.268	
LAB-RMF	0.19	0.24	0.20	0.23	0.21	0.23	0.940	
RAB-LMF	0.14	0.27	0.17	0.26	0.21	0.22	0.354	
RAB-RMF	0.13	0.29	0.08	0.27	0.20	0.28	0.115	
LMF-RMF	0.68	0.33	0.63	0.42	0.60	0.38	0.072	

Abbreviations: LRF, left rectus femoris; RRF, right rectus femoris; LBF, left biceps femoris; RBF, right biceps femoris; LAB, left rectus abdominus; RAB, rectus abdominus; LMF, left multifidus; RMF, right multifidus. Note on shading: White shading = trivial association, light gray shading = weak association, darker gray shading = moderate association.

## Data Availability

The data and code that support the findings of this study are available from the corresponding author, J.M, upon reasonable request.

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
