# Peer review of "Effects of Law Enforcement Load Carriage Systems on Muscle Activity and Coordination during Walking: An Exploratory Study"

_sensors, 2023, doi:10.3390/s23084052_

Round 1

Reviewer 1 Report

This is an interesting paper looking a low back pain indicators through EMG for law enforcement equipment. Overall the study is good and provides some interesting data. My main criticism  would be that the trial only lasted for 60 seconds - longer trial time even with short data capture would have been a better methodology and this should be mentioned in the discussion. However, the study provides some interesting results in an area where there is previously no data.

Other specific comments/suggestions would be:

* Abstract - what are TV and DB - please define

* Abstract line 18 - what were the values of correlations

* Line 19 - how small is small - please state

* line 64 - suggest change "several" to "a few"

* line 99 can you be clear if the data was taken from 24 participants who were LEO or if they were not. This would mean they were either used to the LC or not.

* line 129 - 60 seconds is quite a short walking period. There would be no fatigue and this is not indicative of LEOs working 8 hr shifts. Just a comment but for further work would suggest walking for longer period (maybe 15 - 20 minutes) even if only taking data for short 30 sec period

* line 164 - normally you would give city and country of company  - this may not be the convention in sensors journal - please correct if it is (this might be relevant for other equipment throughout)

* Figure 1 - shows good evidence that there are differences in EMG for different conditions. Is it possible to explain in the figure title if this is an example plot for one subject or if this is mean data across all subjects? It would just make this clearer to the reader to state this in the title

* line 186 - can you be clearer to say "values are median and interquartile range in the bracket."

* line 209 - good that you have said that this is a short exposure. Do you hypothesise that a longer duration activity would have a great effect? If so please do tell the reader. Also would this be part of your suggested future work to have longer durations? If so again please tell the reader

* line 225 - did these previous studies only have short durations for activities like your study? Or did they have longer durations of activity? Could this explain the lower magnitudes that you observed? Again please explain more for the reader about these points

* line 236-237 - can you add a sentence to link core stability to muscle coordination for clarity for the reader.

* line 251 - please also mention that your subjects only performed 60 sec of activity. This is another limitation of your study as longer durations of activity would be more representative of work place activity experienced by LEO

* line 263 - this could combine my previous comment

* line 268 - can you add "LC" to this sentence to make it clear you are talking about this condition for readers who might just skip to your conclusion

Author Response

Reviewer #1:

This is an interesting paper looking a low back pain indicators through EMG for law enforcement equipment. Overall the study is good and provides some interesting data. My main criticism  would be that the trial only lasted for 60 seconds - longer trial time even with short data capture would have been a better methodology and this should be mentioned in the discussion. However, the study provides some interesting results in an area where there is previously no data.

Response: Thank you for your time reviewing our paper and the constructive feedback provided. We agree that longer duration wear of the load carriage systems would be valuable and our lab hopes to address this in the future. The present study was developed based on the question that comparing the types of load carriage on the biomechanical parameters has not been reported to our knowledge. Thus our study was exploratory in nature with desired outcomes of producing preliminary data that would inform future research.

Other specific comments/suggestions would be:

* Abstract - what are TV and DB - please define

Response: We had defined in the first sentence but upon review prompted by the reviewer comment have decided to no longer use these abbreviations in the abstract to minimize abbreviations. As some readers may only read our abstract, it may improve the readability and clarity.  

* Abstract line 18 - what were the values of correlations

Response: The range of values of correlations have been added for each pair of muscles across the 3 conditions.

* Line 19 - how small is small - please state

Response: We’ve added some specific details regarding the partial eta squared values to show the magnitude of the effects and indicated that p-values were less than 0.05 to emphasize they were statistically significant yet generally small in magnitude.

* line 64 - suggest change "several" to "a few"

Response: The edit has been made.

* line 99 can you be clear if the data was taken from 24 participants who were LEO or if they were not. This would mean they were either used to the LC or not.

Response: We have added some text to clarify they were not LEO. This is a good point for the reviewer to raise and a reason why the first limitation mentioned was that the participants were not LEO. Future studies should include LEO as participants.

* line 129 - 60 seconds is quite a short walking period. There would be no fatigue and this is not indicative of LEOs working 8 hr shifts. Just a comment but for further work would suggest walking for longer period (maybe 15 - 20 minutes) even if only taking data for short 30 sec period

Response: Thank you for the comment and we fully agree with the reviewer. We will say that following this study we were curious as to whether an 8-hour duration would result in systematic changes in muscle activity (increase or decrease). From quickly pilot testing 3 subjects, we did not notice a consistent pattern of either increasing or decreasing muscle activity. Nonetheless, an adequately powered study in terms of numbers of participants should be conducted to provide evidence either way on the effects of LEO LC over longer durations.

* line 164 - normally you would give city and country of company  - this may not be the convention in sensors journal - please correct if it is (this might be relevant for other equipment throughout)

Response: Thank you for the feedback. We checked a number of recent articles in Sensors and the reviewer is correct that the country should be given. For equipment and software in the USA typically the state is provided as well. We’ve made multiple edits to the methods to reflect this feedback.

* Figure 1 - shows good evidence that there are differences in EMG for different conditions. Is it possible to explain in the figure title if this is an example plot for one subject or if this is mean data across all subjects? It would just make this clearer to the reader to state this in the title

Response: We definitely can make this edit and agree that adding description will improve the clarity of the figure. As opposed to editing the title, we have clarified in the note below the figure. We hope the reviewer finds this to be acceptable

* line 186 - can you be clearer to say "values are median and interquartile range in the bracket."

Response: This edit has been made. Thank you.

* line 209 - good that you have said that this is a short exposure. Do you hypothesise that a longer duration activity would have a great effect? If so please do tell the reader. Also would this be part of your suggested future work to have longer durations? If so again please tell the reader

Response: We would hypothesize that longer duration activity would have a greater effect. Particularly in individuals with lower levels of muscle strength and endurance in the core musculature. In hindsight, we regret not collecting some basic fitness measures on the participants to profile aspects of fitness that could contribute to individual differences in response to the load carriage conditions. We’ve tried to clearly state the shortcomings of 1) non-LEO population in the study, 2) lack of fitness assessment data and 3) short duration of the trials and overall study in the limitation. Please let us know if you believe we can revise to improve the clarity of these issues.

* line 225 - did these previous studies only have short durations for activities like your study? Or did they have longer durations of activity? Could this explain the lower magnitudes that you observed? Again please explain more for the reader about these points

Response: The durations of trials in previous studies referenced are provided below. Standing trials did use very short durations (30 s). Two of the walking trials (Rice et al. and Simpson et al.) use relatively long durations (>1 hour) and one used 3 to 5 minute trials (Silder et al.). The durations surely could contribute but unfortunately the walking trials did not record from muscles of the abdominals or low back. Interestingly, Simpson et al. found that increased walking distance decrease the amplitude of several muscles in the lower extremity. However, Rice and colleagues found no difference in the amplitude of muscle activity. Silder et al looked at the effect of various loads on muscle activity but not whether activity levels changed over the trials as they were fairly short. We’ve described these findings in the limitations in relation to the recommendation that future research use longer duration trials.

Standing Trials: Motmans et al. and Park et al. both used trials 30 seconds in duration.

Motmans, R.R.E.E.; Tomlow, S.; Vissers, D. Trunk Muscle Activity in Different Modes of Carrying Schoolbags. Ergonomics 2006, 49, 127–138, doi:10.1080/00140130500435066.

Park, H.; Branson, D.; Kim, S.; Warren, A.; Jacobson, B.; Petrova, A.; Peksoz, S.; Kamenidis, P. Effect of Armor and Carrying Load on Body Balance and Leg Muscle Function. Gait Posture 2014, 39, 430–435, doi:10.1016/j.gaitpost.2013.08.018.

Walking Trials

12.8 km distance which took 150 minutes on average.

Rice, H.; Fallowfield, J.; Allsopp, A.; Dixon, S. Influence of a 12.8-Km Military Load Carriage Activity on Lower Limb Gait Mechanics and Muscle Activity. Ergonomics 2017, 60, 649–656, doi:10.1080/00140139.2016.1206624.

8 km distance

Simpson, K.M.; Munro, B.J.; Steele, J.R. Backpack Load Affects Lower Limb Muscle Activity Patterns of Female Hikers during Prolonged Load Carriage. J. Electromyogr. Kinesiol. Off. J. Int. Soc. Electrophysiol. Kinesiol. 2011, 21, 782–788, doi:10.1016/j.jelekin.2011.05.012.

3 to 5 minute trials

Silder, A.; Delp, S.L.; Besier, T. Men and Women Adopt Similar Walking Mechanics and Muscle Activation Patterns during Load Carriage. J. Biomech. 2013, 46, 2522–2528, doi:10.1016/j.jbiomech.2013.06.020.

* line 236-237 - can you add a sentence to link core stability to muscle coordination for clarity for the reader.

Response: This is a good point by the reviewer. We have added a sentence to improve the clarity and connection between core stability and muscular coordination. A citation was added to support the sentence.

McGill, S.M.; Grenier, S.; Kavcic, N.; Cholewicki, J. Coordination of Muscle Activity to Assure Stability of the Lumbar Spine. J Electromyogr Kinesiol 2003, 13, 353–359, doi:10.1016/s1050-6411(03)00043-9.

* line 251 - please also mention that your subjects only performed 60 sec of activity. This is another limitation of your study as longer durations of activity would be more representative of work place activity experienced by LEO

* line 263 - this could combine my previous comment

Response: We completely agree with the reviewer about duration of the study and longer durations would better simulate actual work conditions and could elicit more interesting findings. We've made some edits to the limitations on this issue.

* line 268 - can you add "LC" to this sentence to make it clear you are talking about this condition for readers who might just skip to your conclusion

Response: The edit has been made to clarify we are referring to the LC condition.

Reviewer 2 Report

This study discussed about the effect of load carriage system on the muscle activity and coordination during walking. The manuscript is well written and interesting for biomechanics researchers. However, some information should be added to the manuscript before it can be accepted for publication. My comments are as follows.

1.      Please add a figure showing LEO belt and LEO tactical style weight vest. Thus, readers can understand the test conditions.

2.      Please add a figure showing the locations of sEMG sensor on subjects’ body.

3.      Please add a graph showing a representative time history result of the sEMG signal during the gait task.

4.      It is difficult to understand the method used for data analysis (2.3 Statistical Analysis). During a gait task, several extreme values could be obtained. Did you calculate an average value of those extreme values? Does Figure 1 show the average value?  

5.      Note in Figure 1 mentions that the values are medians. However, the title of section 3.1 explain it as means. Please check.

6.      Just a comment. As mentioned in limitation, the duration of the task is too short. Muscle fatigue should be considered in future study.

Author Response

Reviewer #2:

This study discussed about the effect of load carriage system on the muscle activity and coordination during walking. The manuscript is well written and interesting for biomechanics researchers. However, some information should be added to the manuscript before it can be accepted for publication. My comments are as follows.

 Response: Thank you for your time reviewing our paper and the helpful comments provided.

  1. Please add a figure showing LEO belt and LEO tactical style weight vest. Thus, readers can understand the test conditions.

Response: Thank you for the suggestion. We’ve added a figure to show the LEO belt and Tactical style vest. This is now figure 1.

  1. Please add a figure showing the locations of sEMG sensor on subjects’ body.

Response: We’ve added a figure to show the locations of the sEMG sensor on the subjects body. This is now figure 2.

  1. Please add a graph showing a representative time history result of the sEMG signal during the gait task.

Response: We’ve added a figure to show a representative time history result of the sEMG signal during the gait task. This is now figure 3. As a note we decided to only include 10 s as longer durations resulted in a graph that become much more difficult to visualize due to the high number of gait cycles in the 30 s data collection window.

  1. It is difficult to understand the method used for data analysis (2.3 Statistical Analysis). During a gait task, several extreme values could be obtained. Did you calculate an average value of those extreme values? Does Figure 1 show the average value?

Response: We first computed mean muscular activity values across individual walking trials per subject. For example, for muscle activity, we computed mean activity levels for each of the 8 muscles while walking for each condition and for each subject. Then, to properly compare between conditions, we normalized activity during the belt and vest condition to the unloaded control condition. We did this because we were interested in comparing the load conditions to the unloaded control and to each other. Additionally, expressing results as relative to the control condition allowed us to compare across individuals who had much greater or lower absolute values of muscle activity. This was the approach used by Lindner and colleagues we cited on what is now line 148. Then, within subjects, we did identify some extreme values per condition (belt or vest). These values were winsorized to 3SD and described on line 193 in revised manuscript. Figure 4 (formerly the figure 1 referenced) shows the median and interquartile range across subjects by load conditions. We’ve made a few edits to better describe the statistical procedures in order of operations for the readers. Please inform us if there is more information we can provide to clarify.

  1. Note in Figure 1 mentions that the values are medians. However, the title of section 3.1 explain it as means. Please check.

Response: Thank you for pointing this out. We removed ‘Mean’ from the section header.

  1. Just a comment. As mentioned in limitation, the duration of the task is too short. Muscle fatigue should be considered in future study.

Response: We agree with the reviewer’s comment, which is also mentioned by the other reviewer. We’ve added a sentence to the limitations to recommend that for studies using longer durations fatigue is quantified rather than assuming longer durations results in fatigue.

Round 2

Reviewer 2 Report

Thank you for your response. It can be accepted for publication now.